# The Participatory Approach in Healthcare Establishments as a Specific French Organizational Model at Hospital Department Level to Prevent Burnout among Caregivers: What Are the Perceptions of Its Implementation and Its Potential Contributions by These Caregivers?

**DOI:** 10.3390/ijerph21070882

**Published:** 2024-07-06

**Authors:** Sophie Arnaudeau, Marion Nickum, Evelyne Fouquereau, Séverine Chevalier, Nicolas Gillet, René Mokounkolo, Julien Lejeune, Romuald Seizeur, Philippe Colombat, Christine Jeoffrion

**Affiliations:** 1PSITEC (Psychology: Interactions Time Emotions Cognition, ULR 4072), University of Lille, F-59000 Lille, France; sophie.arnaudeau@univ-lille.fr (S.A.); marion.nickum@univ-lille.fr (M.N.); 2QualiPsy UR 1901, Department of Psychology, University of Tours, 37000 Tours, France; evelyne.fouquereau@univ-tours.fr (E.F.); severine.chevalier@univ-tours.fr (S.C.); nicolas.gillet@univ-tours.fr (N.G.); rene.mokounkolo@univ-tours.fr (R.M.); lucien.lejeune@univ-tours.fr (J.L.); philippe.colombat@univ-tours.fr (P.C.); 3Institut Universitaire de France (IUF), 75005 Paris, France; 4Neurosurgery Department, Brest University Hospital, University of Brest, LaTIM UMR 1101, 29200 Brest, France; romuald.seizeur@chu-brest.fr; 5LIP/PC2S, University Grenoble Alpes, University Savoie Mont Blanc, 38000 Grenoble, France

**Keywords:** participatory approach in healthcare establishments, quality of work life for caregivers, burnout

## Abstract

(1) Background: Manifestations of burnout are regularly observed in the healthcare population. The participatory approach (PA) is a specific organization in the French health service aimed at preserving and improving the psychological health of these staff at work. The main objective of this study was to explore with healthcare professionals their perceptions of the effectiveness of the four PA components (multi-professional team meetings, in-service training, team support meetings and the project approach) implemented to date within French hospital departments, the methods of their implementation and the potential contributions of such an approach to their quality of working life and working conditions (QWLWC), and the quality of care provided. (2) Methods: Semi-structured qualitative interviews were conducted with 21 healthcare professionals in French hospital departments between March and April 2021. After they were recorded, the collected data was transcribed in full and subjected to thematic analysis. (3) Results: According to care providers, PA is only partially deployed in these departments today. Nevertheless, it is helping to develop multi-professional communication, and improves the quality of life at work as well as quality of care. (4) Conclusions: In the light of these results, the creation of a tool for the large-scale evaluation of PA implementation in hospitals emerges as essential, as its deployment in all hospital departments could help reduce the suffering of care professionals. In addition, a better articulation between the concepts of Magnet Hospitals and those of PA would prove heuristically promising.

## 1. Introduction

Burnout has been studied in many professions, including teaching [1] and the legal profession [2]. Professionals in the healthcare sector are particularly affected [3], notably in oncology, anesthesia and emergency departments, as well as in palliative care services [4] and in care centers for the elderly [5]. Burnout affected almost half of American physicians, even before the coronavirus disease 2019 (COVID-19) pandemic [6]. Moreover, burnout had exponentially increased during the COVID-19 period [7]. In 2022, a cross-sectional study was conducted at a tertiary hospital in which 288 nurses were recruited. The data revealed that 48.6% of the nurses were suffering from burnout. Furthermore, 37.2% exhibited severe emotional exhaustion, severe depersonalization was evaluated in 36.8%, and lastly the results of 46.9% nurses showed low personal accomplishment [8]. An observational study in 2014 assessed severe burnout and a low quality of life in nurses. The results showed that 83 (79%) reported severe burnout and a low quality of life [9]. Thus, a prevalence of burnout in professionals in the healthcare sector up to 50% has been noted in several hospital departments, compared with 5–20% in the general population [5,10,11], which can impact the quality of care and management of patients and their relatives [12,13,14,15,16].

Awareness of the suffering of healthcare professionals on the one hand, and of the link between this alteration in their psychological health and the loss of quality of their services on the other [17,18], led a collective of hematology caregivers united in the Reflection group on support and palliative care in hematology (GRASPH: Groupe de Réflexion sur l’Accompagnement et Soins Palliatifs en Hématologie) to create an innovative organizational model in French and European palliative care in the 1990s: the participatory approach (PA) [19]. Since burnout results in particular from professionals’ poor quality of life at work [20], the aim of this approach is to help caregivers by offering training, meeting places and working groups based on a multidisciplinary approach [21,22,23,24]. PA aims to improve the quality of working life for caregivers and optimize the quality of care for patients and their families [17,19,24]. It has thus become mandatory in all care services for the management of palliative care patients, since the ministerial circular of June 2004 (Circular n° 257 of 9 June 2004; France’s Ministry of Health and Social Protection. http://solidarites-sante.gouv.fr/fichiers/bo/2004/04-28/a0282079.htm (accessed on 2 July 2024)), and this management mode became an accreditation criterion for French short-, medium- and long-stay healthcare establishments in 2008 [25]. The RIST law published in France on 26 April 2021 strongly reaffirms the importance of PA within healthcare structures.

The aim of this article is to explore with healthcare professionals their perceptions of the effectiveness to date of the implementation of the four components of PA (in-house training, team support, multi-professional staff meetings and the project approach) within French hospital departments, the ways in which they have been implemented and the potential benefits of such an approach for their quality of working life and the quality of care provided.

### 1.1. Participatory Approach (PA) and Its Components

PA is divided into four components: multi-professional team meetings, in-service training, team support meetings and the project approach.

The multi-professional team meetings are defined as an ethical, cross-disciplinary exercise, making it possible to define a care project adapted to the complexity of each patient’s situation and to make shared decisions in agreement with the patient and their loved ones [26,27]. Therapeutic proposals aimed primarily at improving the patient’s quality of life, rather than purely technical and medical aspects, are discussed by all professionals to inform decision-making, ensure better understanding and encourage adherence to the project. The quality of exchanges and project development depends on the qualities of the manager (usually a doctor or executive) who leads the meeting [24]. The order in which participants speak must be thought out to encourage genuine information sharing: the caregiver introduces the patient, then the floor is given to the nurse and support care professionals, and lastly to the doctor. The chronology of the meeting should always be the same: round-table for information gathering, synthesis of information, then another round-table for proposals [19,24]. Sharing the knowledge, know-how and information of various team members enhances their diversity and helps them to collaborate with their teammates [28,29]. This sharing is also encouraged by in-house training, the second component of PA.

In-service training concerns all professions and enables teams to get to know each other and helps all those involved to speak up, as well as to convey a common message to share knowledge and enable homogeneous technical learning at team level [17,18,23,24,30]. The training courses are specifically designed to respond to problems encountered by caregivers, and it is the caregivers themselves who propose the topics [22]. The aim is to bring together the department’s various professionals around a single theme, to which each of these professionals may have a different, fragmented approach [31], once again placing the emphasis on overall patient care rather than strictly technical–medical aspects. These training sessions are led by a caregiver or doctor from the department, or even by an outside consultant. The third area of exchange within the PA is team support.

Team support meetings aim to reduce stress and increase coping skills to improve quality of life at work [32] in order to prevent potential burnouts. This team support can take several forms, but they are generally support groups (discussion groups, analysis of practice, debriefing, etc.) or individual support provided by a staff psychologist [27]. The creation of discussion groups often stems from the desire of a team; nevertheless, the agreement of the hierarchy is necessary for their establishment and sustainability [33].

The project approach leads to the setting up of working groups based on the identification of a dysfunction, a need or a desire to improve routine operations [17,18,24,26,30]. The multi-professional working group identifies all the objectives to be achieved, the points to be addressed, and suggests avenues for reflection and proposals for improvement regarding the various aspects identified [26]. This can be a one-off approach based on a specific problem, or a global approach in the form of a service or team project [24,26]. These working groups foster autonomy, empowerment, appreciation and a sense of recognition among caregivers [17,22,27]. They offer employees greater latitude and involvement in the decisions that concern them and contribute to their personal fulfillment. High professional commitment is positively associated with well-being [34] and negatively linked to burnout [2].

### 1.2. Participatory Approach (PA): Quality of Life at Work and Quality of Care

Through the creation of spaces for exchange, PA has a positive influence on caregivers’ quality of life at work [17,22], either directly through staff meetings or training, or indirectly through recognition, communication and collaboration [30]. PA therefore has a positive influence on caregivers’ job satisfaction and commitment [18,24]. The four components analyzed separately appear to play a role in this link between PA and quality of life at work [17,18]. By enabling listening and communication, these places of exchange provide team members with recognition from their peers and hierarchy [22], which is essential, as a lack of recognition is the leading cause of work-related suffering among caregivers [30]. These team discussions, during which caregivers talk about the difficulties associated with their practice, could even be decisive for their well-being and mental health. According to Estryn-Bahar [35], giving one’s opinion, receiving advice from others and receiving support from colleagues and superiors are factors in job satisfaction and professional fulfillment. What is more, these times of exchange could represent a form of interpersonal emotional regulation and have a beneficial effect on burnout prevention. They are therefore essential operators in terms of both health and quality of life at work [36,37]. Burnout can indeed be the result of occupational stress or significant relational difficulties at work [3,38,39]. Moreover, workload increases the risk of burnout, while control capacity reduces it [2]. PA thus promotes the prevention of both occupational stress and, ultimately, burnout through involvement in decision-making and the valuing of work by both the hierarchy and patients [23]. The psychic cost of burnout is particularly high [40]. Caregiver burnout generates emotional detachment from the patient, negative attitudes towards the patient and a lack of compassion [11,17,41]. Aggressive behaviors are then more frequent in burnt-out caregivers, and an increase in tolerance towards acts of mistreatment is noted [5,11]. High burnout and perceived stress scores within a healthcare team are linked to lower quality and safety of care, as well as lower patient satisfaction [27] and more treatment errors [13].

Supporting caregivers in their missions also means making them more available to accompany and support patients [33]. A link has been observed in oncopediatrics between the implementation of PA and the quality of care for 142 children, based on self-reported measures from children and their parents [24]. A positive correlation between caregivers’ assessment of care quality and that of parents was observed. In addition, certain PA components positively influence certain aspects of the quality of care provided to children: multi-professional team meetings, for example, have a positive influence on children’s and parents’ satisfaction with caregivers’ attitudes. The “team support meetings” component, meanwhile, has a positive influence on the quality of communication with healthcare staff [18,24].

## 2. Context of the Present Research and Objective

The present study is part of a project financed by France’s Cancéropôle Grand Ouest and validated by the Tours-Poitiers Ethics and Research Committee (CER) (CER-TP File n° 12 January 2020). The study’s scientific committee was made up of seven teacher-researchers and two oncology physicians, one of whom initiated the PA within the GRASPH framework.

Recent research on PA [17,18] has already demonstrated its benefits in terms of quality of work life for caregivers and quality of patient care. Nevertheless, although PA has become mandatory in palliative care units and is now an accreditation criterion for French hospitals, we asked ourselves how it was actually implemented within the departments. The aim of this study was therefore to analyze hospital caregivers’ perceptions of: (1) the actual existence of the four components of PA within French hospital departments, and the ways in which PA is implemented, and (2) the potential benefits of this organizational approach within the respondents’ departments.

## 3. Materials and Methods

The COREQ (COnsolidated criteria for REporting Qualitative research) criteria [42] were used to present the various methodological points, research team, study design and analysis of results [43].

### 3.1. Data Collection

The study is based on a series of interviews with caregivers. The interview guide was co-constructed within the research team, then pre-tested with three participants (Cf. Appendix A). It dealt with four dimensions of PA: 1. multi-professional team meetings (e.g., “Are there times when the various professionals come together to discuss care projects and patient management, apart from handovers? Which professionals take part in these meetings?”); 2. in-service training (e.g., “Are any training courses organized within your department? How often is training organized within the department?”); 3. team support meetings (e.g., “Do you have departmental meetings to support caregivers? What are the reasons for these meetings?”); and 4. the project approach (e.g., “Are there any working groups in place to improve department operations and patient care? What do you gain from the project approach?”).

At the start of each interview, participants were told that the data collected would be treated anonymously, that the recordings would be destroyed after transcription, and that the anonymized written data would be archived on the encrypted computer of one of the researchers on the study’s scientific committee. Participants had the option of stopping the interview at any time if they wished. They were interviewed only once. Interviews lasted between 30 min and an hour. Data were collected at the workplace, within the departments. During the interviews, only the interviewee and the researcher conducting the interview were present. All interviews were recorded and fully transcribed.

### 3.2. Participants

Participants were recruited by convenience from four hospitals and seven departments identified as having a priori implemented a PA: hematology, pediatric onco-hematology, medical oncology, post-emergency internal medicine, pediatric oncology, pediatric onco-hemato-immunology and acute hospitalization. There were three participants in each of the departments investigated. Participants were selected according to their profession. A total of 21 interviews were conducted by one of the researchers on the research team. The sample was made up of seven nurses, seven health managers, five care assistants and two nursery assistants. The vast majority were women (*n* = 20), the average age was 38 (SD = 9.6), and the average number of years of professional experience was 10.5 (M = 10.5; SD = 6.6).

### 3.3. Data Analysis

After transcribing the interviews, we conducted a thematic analysis. This method refers to the transposition of a given corpus into a number of themes representative of the content being analyzed, in relation to the research focus [44]. Participants’ discourse was categorized according to the four dimensions of PA. Coding was performed independently by two researchers who then compared results within each sub-theme and category. The choice of sub-themes, categories and their interpretation were discussed by the two coders and a third researcher. We carefully highlighted participants’ discourse by ensuring that each theme was illustrated with relevant verbatim quotes. We finally discussed these results within our scientific committee.

For each dimension, we begin with a general presentation of the perception of its implementation, based on the verbatim collected, as well as the limitations mentioned by the participants. Secondly, a table lists the sub-themes and the categories to which they refer, followed by examples of quotes (“verbatims”). Verbatims are identified by the participant’s number (from P1 to P21).

## 4. Results

### 4.1. Theme 1: Multi-Professional Team Meetings

Multi-professional team meetings are the aspect that generated the most verbatim comments. According to the carers interviewed, they have been set up in almost all the departments surveyed, generally once a week, and are well described as multidisciplinary, with the presence of carers, managers, support-care professionals and students. In the event of staff members being unable to attend a multi-professional team meeting, a buddy system is generally set up for each sector. The reasons given for this absence are work overload, being busy with a patient, being on vacation, or being on sick leave. Staff meetings are often managed by the doctor, and sometimes by the health executive. According to our interviews, it is often the care assistant who introduces patients, but when this is not the case, the intern, nurse or doctor takes care of introducing them. The caregivers interviewed in our study all felt that their opinions were taken into account; however, they did not always feel that consensus was sought, depending on the type of decision to be made. Consequently, there is a condition for setting up multi-professional team meetings that does not seem to be respected in the departments surveyed. Doctors often remain the decision-makers, particularly when it comes to medical decisions, sometimes seeking the opinion of nurses and support care professionals (Cf. Table 1).

### 4.2. Theme 2: In-Service Training

In all the departments surveyed, in-service training is organized every month or two. Most of the time, caregivers are informed by bulletin boards and emails. For the majority of caregivers, training hours are always as convenient as possible, usually at shift changeover time (2:30 p.m.). Caregivers feel that everything is conducted at organizational level to ensure that they can attend. For example, videoconferencing access is sometimes possible, or training can be duplicated. In the departments we interviewed during our study, the themes of in-service training were mostly related to the pathologies encountered in the department. However, contrary to the recommendations of the PA designers, the choice of themes is generally made by management or doctors, and not by the teams. Similarly, training courses do not necessarily involve all socio-professional categories. Sometimes, all the department’s professionals are present, but in some departments, only the paramedical team attends. Finally, in a few rare cases, the presence of the manager, social worker and psychologist is emphasized, depending on the pathologies being treated.

Finally, for the caregivers interviewed, in-service training often takes the form of a lecture, delivered in most cases by the doctor who is the referent for a particular area. Although not always in a participative format, participants nevertheless say they have the opportunity to ask questions, exchange views and interact with the trainer (Cf. Table 2).

### 4.3. Theme 3: Team Support Meetings

In the departments we surveyed, there are occasional forms of team support, such as debriefing staff meetings. These meetings can be set up after a situation that has been complicated and difficult for the team to deal with, given the difficulty of the treatment, or for an emotionally charged situation (e.g., a death). It is often the health manager who suggests that the team organize meetings subsequent to these complex situations, on a voluntary basis, with a psychologist from outside the department. However, this is not always conducted in crisis situations. On a regular basis, a discussion group is often set up to enable caregivers to talk about care practice in general (Cf. Table 3).

### 4.4. Theme 4: Project Approach

In the departments surveyed, the project approach is fairly well developed. Caregivers participate on a voluntary basis. However, the majority of working groups are composed solely of nurses and care assistants. More rarely, some working groups also include a doctor, a psychologist and/or a health manager. Doctors are often the working group’s referent, but in general, the manager supervises and coordinates. The choice of themes often stems from an observation made by the nursing team and is then discussed by the team as a whole. Examples include the use of a virtual reality helmet, the hygiene group, the pain group, and department decoration projects. Group work times are mostly organized at 2:30 p.m., at shift changeover time. The length of time over which projects are spread out varies considerably according to the different themes, with an average of two years. The caregivers interviewed always express the feeling that consensus is sought, that their opinions are considered in working groups and that decisions are implemented (Cf. Table 4).

## 5. Discussion

The aim of this study was to examine healthcare professionals’ perceptions of the effectiveness of PA and the ways in which it has been implemented in French hospital departments. The aim was also to gather these same professionals’ perceptions of the potential benefits of such an approach within their departments.

### 5.1. Main Contributions

To our knowledge, this qualitative study is the first in-depth exploration of the PA. By analyzing in detail the implementation of PA dimensions as described in the literature, our interviews enabled us to show that PA was partially present in the majority of the departments investigated. Indeed, not all the dimensions were reported in all the departments where interviews were conducted: multi-professional team meetings were in the majority, while in-service training was in the minority, and team support meetings and the project approach were even more rarely present. Moreover, even if the four dimensions of PA were “present” within the departments, the participants in this study reported that some practices did not meet PA criteria. In multi-professional staff meetings, the order of going round the table was not always respected; for example, the care assistant was not automatically the person to introduce patients. Also, decision-making was not systematically collegial and was more the responsibility of the doctors. However, all healthcare professionals associated multi-professional staff meetings with greater communication between professionals and improved patient care.

During in-service training sessions, not all professions were involved, contrary to the framework defined by the PA. Moreover, the professionals we interviewed indicated that most of the training topics were chosen by doctors, and that these courses were more like lectures given by doctors, rather than multi-professional exchanges. In the end, however, the caregivers noted that these training courses were of great benefit, providing theoretical input on pathologies and medications, as well as exchanges relating to their experience within the department—in short, real personal and professional enrichment. Indeed, all participants indicated that these moments enabled them to improve patient care. Videoconferencing access to courses makes them available to as many people as possible. It would be important to standardize this access within teams.

With regard to team support, special meetings following an experience that could have an impact on the team (such as the death of a patient) were not always set up. The importance of informal support among members of the care team, particularly between care assistants and nurses, can nevertheless be underlined. This could take the form of sharing experiences with a colleague in the break room, at mealtimes or during transmissions. However, where formal team support sessions existed within the departments, caregivers reported greater understanding of others (colleagues, patients, families) and improved quality of care.

On the subject of the project approach, we noted several important team projects, such as one aimed at developing virtual reality in a pediatric oncology department. However, in the majority of cases, not all professions were involved, and many groups were carried out between nurses and care assistants (and not with support care professionals, health managers and doctors). On the other hand, those responsible for group supervision were often doctors, and more rarely nurses or care assistants. Feedback on group progress to the rest of the team was also rarely mentioned in the services surveyed.

In the end, this qualitative study showed that for the caregivers interviewed there are two major consequences of implementing PA. The first is the quality of work life, which healthcare professionals associate with three of the four dimensions of PA. In line with what has already been emphasized in some studies, e.g., [17,18], caregivers consider that PA promotes a better quality of life and more optimal well-being at work. The second consequence is improved patient care thanks to PA. The caregivers interviewed referred to this optimization of quality of care as a benefit linked to all dimensions of PA, also empirically corroborating the findings of research that has shown that the implementation of PA enables an improvement in the quality of care, through the provision of new knowledge, collective care, ease, and greater understanding of patients [24,33].

Nevertheless, the results of our interviews with healthcare professionals showed that few of them had any precise knowledge of the PA concept, even among the health managers interviewed. After explaining the process, however, the majority of participants felt that PA had been implemented in their hospital services, and expressed their desire to benefit from participative management, even if they did not name it as such.

Thus, our results show that shared decision-making, like PA as a whole, is still not a widespread practice in the healthcare sector. In France, PA is implemented in only 25% to 30% of the departments where it should be [45]. This model triggers resistance due to its cross-functional approach, disrupting hierarchical and authoritarian structures. This leads to a feeling of loss of security and power [30]. The principles of PA are simple, but creating the mindset needed to change mentalities represents a major challenge. Managers, whether caregivers or not, see their authority, linked to their knowledge and decision-making power, called into question. Yet PA requires mutual trust for participants to listen to others and accept to be challenged [23]. On the other hand, those who are traditionally subordinate to them sometimes resist taking on more responsibility, and managers face difficulties linked to resistance to change on the part of certain employees [46].

### 5.2. Limitations and Research Perspectives

This study is based on a sample of healthcare professionals working exclusively in hospitals in western France, since the research was funded by the Cancéropôle Grand Ouest. Nevertheless, it has enabled us to gain a better understanding of the reality of PA in the structures concerned, both from the caregivers’ viewpoint and the benefits that this approach to department management brings to those who benefit from it. Finally, it has already enabled us, on the basis of the interviews conducted, to formalize a series of items for each of the four dimensions of PA, in order to develop an integrative evaluation tool for the implementation of PA, currently being validated in hospitals throughout France.

As the implementation of PA was conceived as a protective factor against burnout in the caregiving population, future studies are essential to assess the strength of the link between the implementation of PA and the reduction in burnout in the caregiving population. To this end, more experimental studies are necessary to examine whether a reduction in caregiver burnout is observed in departments where PA is present compared with those where PA is not implemented. It is also possible that caregiver suffering is underestimated. New conceptions of burnout are emerging, moving away from strict measurement towards diagnosis and viewing burnout as the dynamic network product of symptoms [47]. This would make it possible to study these concepts within the care professions, and thus contribute to better identification and prevention of occupational health issues through an approach to department organization such as PA.

### 5.3. Practical Implications

To disseminate this organizational model, PA training is essential and should be promoted first and foremost to healthcare managers such as doctors, health managers and facility directors [24], and then also to all stakeholders involved through their professional activity in healthcare organizations. Such training would benefit from being integrated at the earliest stages of managers’ careers, within the curricula of medical faculties and hospital management and executive schools. The aim would be to make these professionals aware of the advantages of this type of management, both regarding the quality of the working life of caregivers and their professional commitment and, by extension, the quality of care [30]. Such innovative training has already been set up at the University of Tours but would benefit from being extended throughout France.

## 6. Conclusions

Since 2004, PA has been mandatory for the care of patients in French palliative care services. Nevertheless, all patients with serious illnesses, even in the curative phase, should be able to benefit from it [24,30]. The interest in multi-professional team meetings shown by France’s Institut National du Cancer is an asset for their generalization [24,26]. The aim is to amend the legislation to include all pathologies within these team meetings, particularly oncology, neurology and geriatrics. Developing a life project for such patients requires the collaboration of different caregivers [26,30]. The implementation of PA therefore depends as much on the willingness of caregivers and the commitment of institutions as it does on national health policy bodies [30].

It would be possible to go even further in terms of organizational implications. Indeed, numerous publications have demonstrated that organizational or managerial factors have an impact on the quality of care, e.g., [17]. While PA focuses on a department or a care unit, there is the “magnet” hospital organization model, which is adapted to a facility as a whole. This concept of Magnet Hospitals refers to a voluntary program for hospitals seeking the highest international qualifications in terms of care [47]. Indeed, these establishments show high levels of job satisfaction as well as quality of care [17] and postulate that they go hand in hand [48,49]. Moreover, it is a matter of fostering the desire in caregivers to stay rather than the wish to leave [50]. Indeed, the implementation of Magnet Hospital programs results in low turnover [17,18], effective conflict management [51] and better patient outcomes [17,47]. The model focuses on nurses, their roles, responsibilities and capabilities [17,48]. The aim is to foster the transformation of organizational culture by enabling nurses to participate collectively in decision-making and one way of doing this could be through collective, anonymous surveys. Time dedicated to these questionnaires at the end of meetings could be beneficial for wider participation. Thus, the development of such an organization requires interprofessional support at all levels of the organization [48]. Yet, as we have shown in our study, although the PA’s dialogue spaces and project approach encourage discussion and mutual respect, this requires the active involvement of the manager, who frames interventions, establishes the speaking order, and gives direction [52]. The process aims to achieve consensus, although the final decision may sometimes be taken by managers, doctors or healthcare executives. In fact, an articulation between the principles of PA and those of Magnet Hospitals could prove fruitful both heuristically and for healthcare sectors.

## Figures and Tables

**Table 1 ijerph-21-00882-t001:** Classification of verbatim related to multi-professional team meetings.

Sub-Theme	Categories	Quotes (Verbatims)
Organization	Multidisciplinary of the professionals present	[Staff meetings were attended by] care assistants, the nurses, the psychologist, the dietician, the manager, the social worker, the hotel assistant, the doctors, the interns, the externs, the medical secretary, all the students if there were any, and the department manager (P7).
	Expectations of staff meetings	It gives me a framework for my day, concrete answers to my questions about whether or not to give such and such a drug, and then afterwards, to find out everything about what happens next, how to take care of the patient, to have answers, to know what I’m going to do during the day, whether or not there are any tests to be carried out (P7).
	Decision-making	No, consensus is not sought and […] each patient has a referring doctor, so the doctor can ask for the opinion of the other doctors present, but is often the one who makes the decision (P1).
Contributions	Multiple opinions	Having several viewpoints with different professionals who will not have the same perception (P11).
	Patient information	In an instant, we have a better understanding of patients and their various facets: their situation, their family, external care. It makes the link with the patient’s social history, their day-to-day condition, their needs at home, their life at home, and whether the disease is improving or deteriorating (P6).
Care	Improving care	If we don’t know how to deal with a patient because they have behavioral problems, for example, we may have a colleague who has found a way to communicate with them. We exchange ideas on this, we work in collaboration, so anything that can be useful to the patient’s care, if we can share it, that’s when (P12).
	Action evaluation	If we talked about it the week before, we’ll talk about it again to see if it worked or not, and we’ll evaluate it (P18).
	Anticipation	We look at what’s already in place, and try to think of what else we could put in place if need be (P6).
	Patient outcomes	Planning ahead, what to do after hospitalization, request for palliative care, request for follow-up and rehabilitation care, the return home, under what conditions, with what assistance in place, whether or not the social worker needs to be informed, etc. (P8).
Workgroup	Relations and communication with doctors	Staff meetings will enable direct communication with doctors (P10).
	Information sharing	This enables us to connect the doctors and the paramedical team, so that we’re all working in the same direction and with the same care, because sometimes the doctors have their own point of view about the disease, and we have our own point of view about the patient who’s in difficulty, and we need to be able to agree (P20).
	Team work	It’s really about working together, the collective for the patient (P11).
	Communication with colleagues	Understand colleagues better, be supportive, tell each other things (P6).
Quality of life at work	Support	The multi-professional staff meeting, I think, is a time for exchange and support, because we can talk about anything during this meeting, and it’s true that the basis of our work is the patient, it’s improving their hospitalization conditions, their care, but at the same time, it’s sometimes up to the professionals to express themselves about their difficulties, their feelings… (P2).
	Open expression	Everyone listens to each other, it’s respectful, and we’re not here to judge anyone on the staff (P10).
	Recognition	[care assistant] To feel part of the team, to be listened to, recognized, the importance of the care assistant having their say (P3).
	Well-being at work	At any rate, less suffering (P6).

**Table 2 ijerph-21-00882-t002:** Classification of verbatims relating to in-service training.

Sub-Theme	Categories	Quotes
Satisfaction	Feeling	A very pleasant moment (P16).
	Making sense	The feedback I got from the care assistants was that they understood why they had to monitor urinary pH and all that, some of them did it systematically, because they’d been told they had to, but today, well, they know why they do it and the importance it has on a daily basis in patient care (P2).
	Exchange	It helps communication between professionals (P10).It was basically a lecture, but we have a doctor who was very open to dialogue, so she let us intervene if we had any questions, but it was basically a PowerPoint presentation (P11).
Learning	Professional enrichment	In-service training enables us to work well (P1).
	New insights	It’s good to be always up to date with what’s going on, because it’s true that sometimes there are new treatments, even if new diseases are discovered (P8).
	New practices	Learning new protocols, because they change so often (P5).
	Training time for new recruits	For example, we welcomed a new colleague from another hospital, who didn’t do things the same way (P7).
Relations with patients	Ease of care	Having mastered the latest techniques, we’re more at ease when dealing with patients, and that’s really what it’s all about (P9).
	Improved relations	Understanding a patient’s condition makes it easier to communicate with them (P16).
	Relationship with the family	It helps answer patients’ and families’ questions and fosters a climate of trust (P13).

**Table 3 ijerph-21-00882-t003:** Classification of verbatim related to team support meetings.

Sub-Theme	Categories	Quotes
Quality of work life	Well-being	So, you don’t have to take your worries home with you. Taking a break, discussing difficulties and expressing them in words are a source of well-being (P3).
	Emotional expression	It’s good to let go, to get it off your chest (P19).
Collective	Understanding colleagues	It allows for more exchange, listening and recognition. It allows you to understand your colleagues, what they may be feeling, whether or not it’s the same thing they’re feeling (P6).
	Solidarity and cohesion	Develops strong cohesion within the group, which becomes like a team, paying attention to each other, helping each other out, creating a good atmosphere within the team, knowing that you’re not alone (P16).
	“Corridor support”	Yes, it really allows us to review the situation. We often have a lot of discussions in the corridors between us, we talk a lot about the situation and the health managers decide to set up the meeting to put a stop to the situation (P1).
Care	Opinions on care	It allows us to say things that we might have refrained from saying in front of our department head and health manager (P4).
	Improving care	We try to improve, the aim is always to take the best possible care of patients, and sometimes we realize because of a situation that this wasn’t the case, the aim is also to say to ourselves that next time, we’ll do better (P1).
	Better understanding of situations	It helps us to step back when we’re stuck in a situation, and as a result, we understand better why the child or the family reacts the way they do, and we can better adapt our care with this information (P20).The team was questioning whether we had done the best we could, and we were able to explain. This helped calm the team and give them a better understanding of what had happened in the situation, which was a relief for some people (P6).

**Table 4 ijerph-21-00882-t004:** Classification of verbatim related to the project approach.

Sub-Theme	Categories	Quotes
Quality of life at work	Appraisal	It’s good for everyone, in every case (P9).
	Satisfaction and well-being	Reflect on what we could do better to improve the well-being of caregivers and enhance personal satisfaction with our profession (P17).
	Motivation	Having a recognized position and asserting your ideas make you want to continue down this path, and it’s great to be able to work like that (P16).
	Sense of accomplishment	We bring something extra, we can improve interfaces, visualize improvements, see the progress of projects. You can see that you’re providing answers, and you feel involved in the department (P18).
Collective	Teamwork	It enhances solidarity and allows us to discover each other in a different way. It contributes to collaborative work and team questioning (P7).
	Improving relations	Cohesion, improving relations between different professionals (P18).
	Result of a malfunction identified by the team	Comes from an observation of dysfunction in the care team, stems from concrete, everyday things (P6).
	Recognition	It’s rewarding to be heard, to not just be the little helper doing her little chores (P16).
Care	Improving practices	As close as possible to practices, as useful as possible, help practices evolve, facilitate practices, think about what we could do better for patients, implement what we want in the department, update everything, implement new protocols, always evolve (P17).We’re a very big team, and if we didn’t have all these groups to discuss things with each other, I think everyone would scatter their ideas, so at least here, we’re all united, all doing the same thing (P20).
	Standardizing practices	We noticed that other departments were operating differently, or that practices were evolving, so we realized that perhaps some things were obsolete and needed to be changed (P6).
	Quality of care	When projects are carried out well, it’s positive in terms of what we can offer patients (P6).

## Data Availability

The original contributions presented in the study are included in the article, further inquiries can be directed to the corresponding author.

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
