# Peer review of "The Participatory Approach in Healthcare Establishments as a Specific French Organizational Model at Hospital Department Level to Prevent Burnout among Caregivers: What Are the Perceptions of Its Implementation and Its Potential Contributions by These Caregivers?"

_ijerph, 2024, doi:10.3390/ijerph21070882_

Round 1
Reviewer 1 Report
Comments and Suggestions for Authors
Thank you for inviting me to review.
Introduction: Please conduct some additional wider literature review on burnout, especially among medical professionals.
Methods: Please also include the original reference for COREQ criteria - Tong et al. 2007, as well.
- How were the participants chosen from each of the departments investigated? random vs. convenience
Overall, well written article.
Author Response
We thank you for all your advices. You will find below our responses to each comment and question. We have modified the manuscript on the basis of your feedback and suggestions, and hope that this revised version of the article will meet with your approval.
Responses to comments are written in italics in this letter and in red in the manuscript. We remain at your disposal should you require any further information.
With our thanks and best regards.
Reviewer 1:
« Thank you for inviting me to review
- Introduction: Please conduct some additional wider literature review on burnout, especially among medical professionals.
We have added a reference concerning burnout « Tharani et al., 2023; Squellati & Zangaro, 2022 » in red color, on page 4, line 155 and line 156. « Tharani et al., 2023 » in red color, on page 2, line 48 and in the bibliography, and « Greep et al., 2022; Koenig, 2023; Tawfik et al., 2018 » in red color, on page 2, line 64. We have added a study: « Burnout affected almost half of American physicians, even before the coronavirus disease 2019 (COVID-19) pandemic (Shanafelt et al., 2015) ». in red color, on page 2, line 50 and in bibliography. We have also added a reference and a study concerning nurse burnout “Moreover, burnout had exponentially increased during the Covid-19 period (Sullivan et al., 2022). In 2022, a cross-sectional study was conducted in a tertiary hospital in which 288 nurses were recruited. The data revealed that 48.6% of nurses suffered from burnout. 37.2%, exhibited severe emotional exhaustion, severe depersonalization was evaluated in 36.8%, and lastly the results of 46.9% nurses showed low personal accomplishment (Andlib et al., 2022) » on page 2, line 53 and the reference in bibliography. We have also added another study “An observational study in 2014 assessed severe burnout and a low quality of life in nurses. The results showed that 83 (79%) reported severe burnout and a low quality of life (Naz et al., 2016). “ on page 2, line 57, and in bibliography in red color.
- Methods: Please also include the original reference for COREQ criteria – Tong et al., 2007.
We have added this reference in red on page 4, line 199 and in the bibliography.
- How were the participants chosen from each of the departments investigated? random vs convenience.
We have added this clarification in the manuscript on page 5, line 223. « Participants were recruited by convenience ».
Overall, well written article. »
We thank you for your feedback and suggestions.
Reviewer 2 Report
Comments and Suggestions for Authors
The authors present an important aspect of organizational decision-making, and point out the many reasons why the implementation fails. These are primarily linked to human-factors, making the problem even more challenging to implement and impactful at larger scale. Similar problems are seen in industrial sectors, where the organizational culture affects the enterprise-wide decision making.
Few comments/questions for the authors to add in the manuscript Discussion or Conclusion section:
1. How can the use of technology ensure a more inclusive and useful team interaction? Can use of live anonymous polls during meetings help overcome the authority/hierarchy barrier?
2. How does one design these questionnaires for large-scale user input and automated feedback and analysis? A more quantitative analysis is required to research these problems further.
Author Response
We thank you for all your advices. You will find below our responses to each comment and question. We have modified the manuscript on the basis of your feedback and suggestions, and hope that this revised version of the article will meet with your approval.
Responses to comments are written in italics in this letter and in red in the manuscript. We remain at your disposal should you require any further information.
With our thanks and best regards.
Reviewer 2:
The authors present an important aspect of organizational decision-making, and point out the many reasons why the implementation fails. There are primarily linked to human-factors, making the problem even more challenging to implement and impactful at larger scale. Similar problems are seen in industrial sectors, where the organizational culture affects the enterprise-wide decision making.
Few comments/questions for the authors to add in the manuscript Discussion or Conclusion section:
- How can the use of technology ensure a more inclusive and useful team interaction? Can use of live anonymous polls during meetings help overcome the authority/hierarchy barrier?
Technology is sometimes used for in-house training. Access to courses is sometimes possible via videoconferencing, enabling slideshows to be broadcast. This enables more useful team interaction, as everyone can access in-house training, and team interaction is therefore based on the same knowledge. It would be important for this videoconferencing access to be standardized so that as many people as possible can have access to it. We have added to the manuscript under discussion: « Videoconferencing access to courses makes them available to as many people as possible. It would be important to standardize this access within teams. » in red color, on page 12, line 340. What's more, the use of surveys could effectively enable more collective decision-making, and anonymity would allow employees to express themselves more freely. We've added this to the manuscript, in the conclusion section: « and one way of doing this could be through collective, anonymous surveys » in red color, on page 14, line 446.
- How does one design these questionnaires for large-scale user input and automated feedback and analysis? A more quantitative analysis is required to research these problems further.
We would like to thank you for your very interesting comment. Dedicated time for these questionnaires would enable wider participation and more quantitative analyses. We have added to the manuscript, in the conclusion section « Time dedicated to these questionnaires at the end of meetings could be beneficial for wider participation. » in red color, page 14, on line 447.
Reviewer 3 Report
Comments and Suggestions for Authors
I have read the submitted study and overall it constitutes a very good quality qualitative study but there are a couple of issues that need to be addressed:
-The paper provides a detailed description of the context of the study in France but offers little insights on how the scope and results of their study compare with other studies that share similarities with the PA approach in other contexts. Both the introduction to the study and its discussion will be benefited from those insights.
-The authors explain that they applied a thematic content analysis but they provide no references to their approach and no specific details on the specific process. What were the main characteristics of their coding process? Did they use any software?
-Were there any noticeable differences within their analysis sections and subsections across the different professional groups?
-Was there any difference between more experienced and less experienced caregivers?
Author Response
We thank you for all your advices. You will find below our responses to each comment and question. We have modified the manuscript on the basis of your feedback and suggestions, and hope that this revised version of the article will meet with your approval.
Responses to comments are written in italics in this letter and in red in the manuscript. We remain at your disposal should you require any further information.
With our thanks and best regards.
Reviewer 3:
I have read the submitted study and overall it constitutes a very good quality qualitative study but there are a couple of issues that need to be adressed:
- The paper provides a detailed description of the context of the study in France but offers little insights on how the scope and results of their study compare with other studies that share similarities with the PA approach in other contexts. Both the introduction to the study and its discussion will be benefited from the insights.
We thank you for your comments. The participatory approach is a French one. However, we mention the case of Magnets Hospitals in the USA, which show similarities with the participative approach in the conclusion: « While PA focuses on a department or a care unit, there is the “magnet” hospital organi-zation model, which is adapted to a facility as a whole. This concept of Magnet Hospitals refers to a voluntary program for hospitals seeking the highest international qualifications in terms of care (Friese et al., 2015). Indeed, these establishments show high levels of job satisfaction as well as quality of care (Lejeune et al., 2020), and postulate that they go hand in hand (Anderson et al., 2018; Friese et al., 2015). Moreover, it is a matter of foster-ing the desire in caregivers to stay rather than the wish to leave (Sibé & Alis, 2016). In-deed, the implementation of Magnet Hospital programs results in low turnover (Lejeune et al., 2020, 2021), effective conflict management (Cannasse, 2008) and better patient out-comes (Friese et al., 2015; Lejeune et al., 2020). The model focuses on nurses, their roles, responsibilities and capabilities (Anderson et al., 2018; Lejeune et al., 2020). » page 13, line 433.
- The authors explain that they applied a thematic content analysis but they provide no references to their approach and no specific details on the specific process. What where the main characteristics of their coding process? Did they use any software?
The main characteristics of our coding are the implementation (or not) and the contributions, benefits and limits of each of the components of the participatory approach. We have provided the missing details on the coding method used as indicated in the “data analysis” section, page 5, line 234: “After transcribing the interviews, we conducted a thematic analysis. This method refers to the transposition of a given corpus into a number of themes representative of the content being analyzed, in relation to the research focus (Paillé & Mucchielli, 2008). Participants’ discourse was categorized according to the four dimensions of PA. Coding was performed independently by two researchers who then compared results within each sub-theme and category. The choice of sub-themes, categories and their interpretation were discussed by the two coders and a third researcher. We carefully highlighted participants’ discourse by ensuring that each theme was illustrated with relevant verbatim quotes. We finally discussed these results within our scientific committee.” We did not use any software.
- Were they any noticeable differences within their analysis sections and subsections across the different professional groups?
We found no major differences between the different professional groups. Indeed, on the four dimensions of the PA, the three professional categories gave similar answers and had equal representations. For future studies, it would seem appropriate to re-examine this inter-professional difference by including other caregivers (e.g. doctors) or between services.
- Was there any difference between more experienced and less experienced caregivers?
We did not conduct comparative analyses of responses between the most and least experienced caregivers. This was not part of our research hypotheses.